# Can syndromic surveillance help forecast winter hospital bed pressures in England?

Roger A. Morbey[1]*, Andre Charlett[2], Iain Lake[3], James Mapstone[4], Richard Pebody[5], James Sedgwick[6], Gillian E. Smith[7], Alex J. Elliot[7]

1 National Infection Service, Public Health England, Birmingham, England, United Kingdom, 2 Department Head, Statistics and Modelling Economics Department, Public Health England, London, England, United Kingdom, 3 School of Environmental Sciences, University of East Anglia, Norwich, England, United Kingdom, 4 Public Health England, Bristol, England, United Kingdom, 5 National Infection Service, Public Health England, London, England, United Kingdom, 6 National Infection Service, Public Health England, Ashford, England, United Kingdom, 7 National Infection Service, Public Health England, Birmingham, England, United Kingdom

* roger.morbey@phe.gov.uk

## Abstract

### Background

Health care planners need to predict demand for hospital beds to avoid deterioration in health care. Seasonal demand can be affected by respiratory illnesses which in England are monitored using syndromic surveillance systems. Therefore, we investigated the relationship between syndromic data and daily emergency hospital admissions.

### Methods

We compared the timing of peaks in syndromic respiratory indicators and emergency hospital admissions, between 2013 and 2018. Furthermore, we created forecasts for daily admissions and investigated their accuracy when real-time syndromic data were included.

### Results

We found that syndromic indicators were sensitive to changes in the timing of peaks in seasonal disease, especially influenza. However, each year, peak demand for hospital beds occurred on either 29th or 30th December, irrespective of the timing of syndromic peaks. Most forecast models using syndromic indicators explained over 70% of the seasonal variation in admissions (adjusted R square value). Forecast errors were reduced when syndromic data were included. For example, peak admissions for December 2014 and 2017 were underestimated when syndromic data were not used in models.

### Conclusion

Due to the lack of variability in the timing of the highest seasonal peak in hospital admissions, syndromic surveillance data do not provide additional early warning of timing. However, during atypical seasons syndromic data did improve the accuracy of forecast intensity.

**Data Availability Statement:** The data underlying the results presented in the study are available from Public Health England, https://www.gov.uk/government/organisations/public-health-england.

**Funding:** The author(s) received no specific funding for this work.

**Competing interests:** The authors have declared that no competing interests exist.

## Introduction

Seasonal variation in emergency hospital admissions causes considerable pressure on health systems. In northern temperate countries, there is a seasonal winter peak in respiratory disease which can contribute to the increasing demand for hospital beds [1]. Importantly, health care planners need to predict the timing and intensity of seasonal demand for hospital beds to make efficient use of resources and avoid any deterioration in health care due to insufficient capacity [2, 3]. Also, increased pressure on services can lead to intensive care unit bed pressures, cancellation of elective surgery and facility closures. Furthermore, past studies have identified atypical seasons when hospital pressures due to influenza are significantly increased [4]. Hospital admissions data on its own may not provide adequate early warning of pressures due to reporting delays. For example, admission data with diagnostic details are not available until after discharge.

Syndromic surveillance involves the real-time monitoring of patients to provide early warning of, for example, seasonal influenza or respiratory syncytial virus (RSV) epidemics. Public Health England (PHE) routinely analyses real-time syndromic surveillance data for potential threats to public health. PHE has four syndromic surveillance systems, designed to monitor patients with morbidity presenting to different parts of the National Health Service (NHS) and provide a comprehensive coverage of patients with varying severity of symptoms. Since its establishment in 2013, PHE has monitored data from NHS 111, a free telephone advice line [5]. Primary care visits to family doctors (called General Practitioners or GPs in the UK) are analysed via two syndromic surveillance systems, one for in-hours (GPIHSS) and one for out-of-hours and unscheduled care (GPOOHSS) [6, 7]. Finally, patients with more severe illness presenting to emergency departments (EDs) are captured via the Emergency Department Syndromic Surveillance System (EDSSS) [8]. These systems are monitored daily with data available today from the previous day's consultations.

Previous studies have shown that general syndromic respiratory indicators (e.g. respiratory tract infections) are associated with the overall burden of seasonal respiratory disease, whilst specific indicators (e.g. influenza-like illness; ILI) can help distinguish between different trends in individual pathogens, e.g. between influenza and RSV [9–11]. Therefore, syndromic surveillance may be able to provide early warning of hospital bed pressures caused by seasonal respiratory disease.

Here, we investigate the relationship between syndromic data and daily emergency hospital admissions. Specifically, we have explored whether daily syndromic data routinely collected for surveillance in England can help predict the timing and intensity of winter peaks in emergency hospital admissions.

## Methods

### Data collection

Respiratory indicators were selected from each of PHE's syndromic systems including general indicators that capture a wide range of acute respiratory illness as well as more specific indicators. The indicators used included: acute respiratory infections, bronchitis, pneumonia and ILI from the ED and GP systems; and cold/flu calls from NHS 111. Each indicator was chosen based upon previously gathered evidence of sensitivity to circulating winter pathogens [9–11]. Syndromic indicators are monitored as daily rates rather than counts because routine surveillance can be affected by daily fluctuations in coverage. For GPIHSS rates are calculated per 100,000 registered patients. In other syndromic systems, registered patient coverage data are not available and rates are calculated using total activity as a denominator (Table 1).

**Table 1. Methods for calculating syndromic indicator rates for syndromic systems.**

| Syndromic Surveillance System | Numerator | Denominator |
|---|---|---|
| GPIHSS | GP in-hours consultations for specific indicator | Registered population of participating GP practices |
| GPOOHSS | GP out-of-hours consultations for specific indicator | Total Read-coded consultations |
| EDSSS | ED attendances for specific indicator | Total ED attendances |
| NHS 111 | Calls for specific indicator | Total calls to NHS 111 |

EDSSS: emergency department syndromic surveillance system; GPIHSS: GP in-hours syndromic surveillance system; GPOOHSS: GP out of hours and unscheduled care syndromic surveillance system; NHS 111: telehealth syndromic surveillance system.

Emergency hospital admission data were extracted from hospital episode statistics (HES), which include all admissions for England [12]. The data were aggregated into daily counts of admissions, using the date of the first consultant episode in each hospital stay. Previous studies in the UK have found that there is a limit to the number of excess admissions, with an artificial ceiling caused by the maximum capacity of hospitals due to a finite number of beds [13]. During extreme busy periods, hospitals are forced to cancel elective admissions in favour of emergency admissions. Therefore, we studied emergency rather than both elective and emergency admissions to give a less biased estimate for surges in demand. We used 'all' emergency admissions in our models rather than 'respiratory' admissions. Although, respiratory admissions were likely to have a stronger association with the syndromic indicators, forecasts for all emergency admissions are more likely to useful to healthcare service planners managing hospital demand.

Data were analysed for winter influenza seasons, between weeks 40 to 20 [14], from October 2013 until February 2018. This period covered the start of historical records for current surveillance systems until the most recent HES data available at the time of analysis. All data were stratified into the 7 age groups used for routine syndromic surveillance: under 1 year, 1 to 4, 5 to 14, 15 to 44, 45 to 64, 65 to 74 and over 75 years. We focussed on daily data routinely used for syndromic surveillance so that any conclusions about associations with hospital admissions would be relevant for using syndromic systems to predict admissions.

There is considerable variation in syndromic data and hospital admissions by day of the week due to availability of services and patient presenting behaviour. For instance, GP in-hours services are not available at weekends or during public holidays and other primary care services are correspondingly busier at these times. However, we did not want to lose resolution by aggregating the data into weekly rates. We want any potential future forecasts developed from syndromic data to be updatable daily. Therefore, we used seven-day moving averages to smooth the data to remove day of the week effects before examining trends or peak timing and included variables for weekends, and public holidays in our regression models.

## Model fitting

The day of peak activity for emergency hospital admissions and syndromic indicators was calculated for each season. In addition, time series were visually inspected to identify seasons with secondary peaks in admissions or syndromic activity.

We used linear regression to test for associations between hospital admissions and syndromic indicators. Models included a variable for linear trend, and a pair of Fourier terms to model seasonal variation [15]. In addition to variables for weekends and public holidays, 25 December

was also included as an additional binomial variable to account for differing presenting behaviour at Christmas compared to other public holidays [16]. Each syndromic indicator was modelled separately, stratified by age group. The adjusted R square of the regression models was used to estimate how much variation in hospital admissions could be attributed to the variables included in each model.

### Model forecasts

We used the same association regression models to assess the utility of syndromic data for forecasting, but with different training data. For these forecasting models, we used k-fold cross validation to simulate the real-world scenario where information on the current season's hospital admissions is not available in real-time [17]. For example, the forecast models for season 2017/2018 were constructed using training data that did not include 2017/2018.

A 'null model' was additionally created to include the same independent variables as the other models but without any syndromic data. Comparison with the null model enabled us to assess the added value of using syndromic data, where yesterday's data is routinely available, to forecast today's admissions, when recent admission data is not assumed to be available. These models were assessed by comparing the average absolute daily forecast errors.

## Results

### Data volume

During the study period (Oct 2013 –Feb 2018), there was a mean of 15,916 hospital admissions each day with 20,002 on the busiest day (29th Dec 2017). The largest syndromic system is GPIHSS where mean coverage was 30.3 million registered patients and a mean count of 22,281 respiratory tract infections each day, with a peak of 62,454. For GPOOHSS the mean of daily consultations with a Read-code recorded was 13,586. For NHS 111 the mean of daily total calls was 30,581. Finally, for EDSSS the mean of total daily attendances from all participating EDs was 5,173.

### Model fitting

In the five seasons studied, peak activity in emergency hospital admissions were noticeably consistent, the all ages peak occurring either on 29th or 30th December every year. There was much more variation in the timing of peaks in syndromic indicators, particularly those most closely associated with influenza activity e.g. ILI. For example, during 2016/17 the peak in ILI consultations occurred during late December/early January, however, in 2015/16 the peak was very late in the season, and did not occur until late March (Fig 1).

All the syndromic indicators studied had significant associations with emergency hospital admissions and most association models explained over 70% of the variation in admissions (Table 2). In general associations were weaker in those aged <15 years, especially for 1–4 year olds. The syndromic indicators with the strongest associations were GPIHSS upper and lower respiratory tract infections for all ages and all age bands except 15 to 44 years.

### Model forecasts

All forecast models used to predict daily admissions showed similar trends and the resulting average absolute errors were low. Daily admissions, averaged around 16,000, but absolute errors were between 14 and 60 (Table 3). Histograms of absolute errors (not shown) were examined, however, there was no evidence of outliers having undue influence. For each age band, the null model was improved by including a syndromic indicator to reduce the forecast

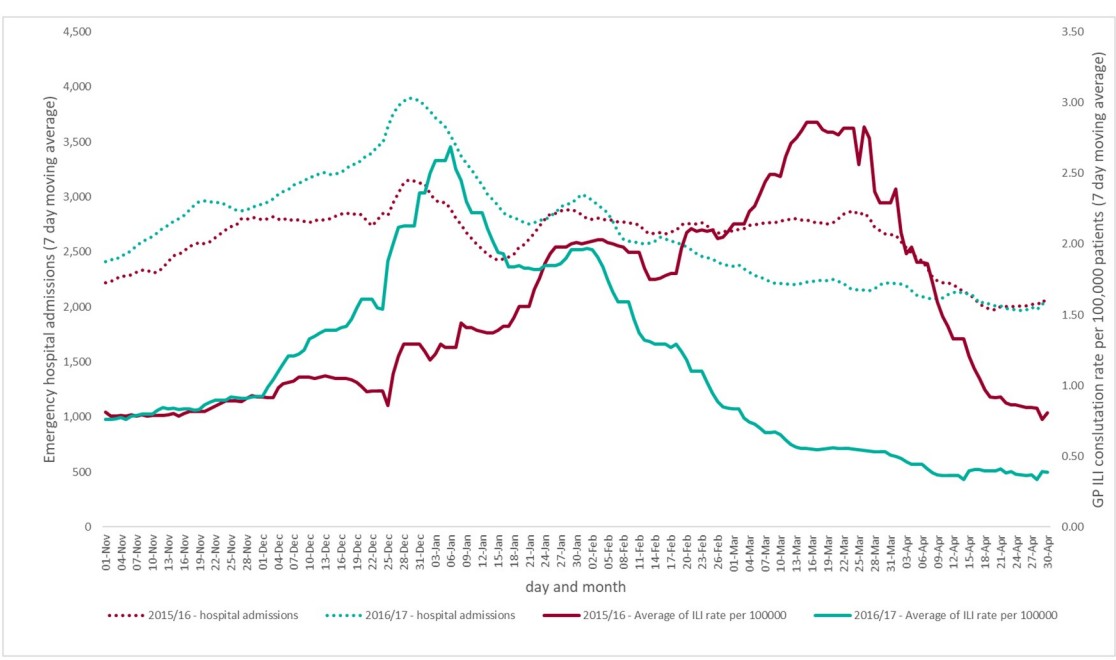

**Fig 1. Comparison of GP influenza-like illness consultations and emergency hospital admissions across two seasons.**

errors. However, for some models the improvement in reduction of forecast errors was marginal (less than two patients a day). The biggest improvements in forecast accuracy were in the youngest and eldest age bands. For example, the average forecast error for under 1s in the null model was 24.12 falling to 14.20 when GP lower respiratory tract infections were

**Table 2. Adjusted R square values (%) for regression models measuring association between syndromic indicators and emergency hospital admissions (higher values are depicted with darker shades–highest value in each column shown in bold).**

| Syndromic system and indicator | Adjusted R square value (%) | | | | | | | |
| --- | --- | --- | --- | --- | --- | --- | --- | --- |
| | Age band (years) | | | | | | | |
| | under 1 | 1 to 4 | 5 to 14 | 15 to 44 | 45 to 64 | 65 to 74 | over 75 | All ages |
| **emergency department syndromic surveillance system** | | | | | | | | |
| acute respiratory infection | 63.7 | 42.03 | 48.46 | 77.88 | 86.14 | 81.25 | 73.38 | 67.55 |
| admitted respiratory | 57.01 | 32.50 | 48.89 | 77.19 | 86.17 | 81.03 | 71.03 | 64.83 |
| bronchitis | 66.03 | 28.82 | 48.37 | 77.22 | 86.16 | 80.93 | 69.64 | 65.31 |
| pneumonia | 47.11 | 26.30 | 48.38 | 77.18 | 86.13 | 81.00 | 70.93 | 62.43 |
| **GP in-hours syndromic surveillance system** | | | | | | | | |
| influenza-like illness | 47.83 | 30.31 | 51.10 | 77.52 | 86.22 | 81.26 | 74.66 | 64.13 |
| lower respiratory tract infection | **78.21** | 51.32 | 49.71 | 77.55 | **86.23** | **82.44** | **77.85** | 71.90 |
| pneumonia | 48.20 | 30.98 | 48.35 | 77.18 | 86.12 | 80.99 | 72.28 | 63.44 |
| upper respiratory tract infection | 71.59 | **59.78** | **55.74** | 77.51 | 86.19 | 82.40 | 77.73 | **72.99** |
| **GP out of hours and unscheduled care syndromic surveillance system** | | | | | | | | |
| acute respiratory tract infection | 67.80 | 46.83 | 48.33 | **78.05** | 86.18 | 82.13 | 76.79 | 69.44 |
| bronchitis | 69.61 | 31.43 | 48.55 | 77.28 | 86.13 | 81.10 | 70.54 | 66.38 |
| **NHS 111– telehealth syndromic surveillance system** | | | | | | | | |
| cold/flu | 44.51 | 29.66 | 49.45 | 78.29 | 86.21 | 81.26 | 74.98 | 63.48 |
| NULL model | 45.61 | 23.89 | 48.31 | 77.15 | 86.12 | 80.86 | 69.13 | 61.58 |

**Table 3. Absolute mean forecast errors (daily admissions) in England by age band (lowest errors in each age band highlighted in bold).**

| Syndromic indicator in model | Absolute mean forecast errors | | | | | | |
|---|---|---|---|---|---|---|---|
| | Age band (years) | | | | | | |
| | under 1 | 1 to 4 | 5 to 14 | 15 to 44 | 45 to 64 | 65 to 74 | over 75 |
| Null model | 24.12 | 27.67 | 19.88 | 45.07 | 35.80 | 26.75 | 54.98 |
| EDSSS: acute respiratory infection | 19.54 | 25.94 | 19.66 | **44.26** | 35.82 | 26.34 | 52.80 |
| EDSSS: admitted respiratory | 21.73 | 26.84 | 19.71 | 45.19 | 36.17 | 27.14 | 59.25 |
| EDSSS: bronchitis | 18.85 | 27.61 | 19.68 | 44.78 | **35.35** | 26.49 | 54.23 |
| EDSSS: pneumonia | 23.28 | 27.04 | 19.70 | 44.72 | 35.50 | 26.54 | 54.04 |
| GPIHSS: lower respiratory tract infection | **14.20** | 21.81 | 19.83 | 45.04 | 35.94 | **25.82** | 46.86 |
| GPIHSS: influenza-like illness | 23.88 | 28.01 | 19.56 | 45.67 | 36.23 | 26.86 | 55.32 |
| GPIHSS: pneumonia | 23.72 | 26.38 | 19.85 | 45.39 | 35.93 | 26.91 | 54.10 |
| GPIHSS: upper respiratory tract infection | 16.68 | **19.63** | **18.75** | 45.13 | 35.97 | **25.82** | **46.68** |
| GPOOHSS: bronchitis | 17.08 | 26.26 | 19.85 | 45.17 | 35.80 | 26.59 | 54.36 |
| GPOOHSS: acute respiratory tract infection | 17.92 | 23.69 | 19.87 | 45.23 | 35.91 | 25.96 | 48.87 |
| NHS 111: cold/flu | 23.51 | 26.88 | 19.31 | 45.56 | 35.39 | 26.09 | 51.67 |

EDSSS: emergency department syndromic surveillance system

GPIHSS: GP in-hours syndromic surveillance system

GPOOHSS: GP out of hours and unscheduled care syndromic surveillance system

NHS 111: telehealth syndromic surveillance system

included in the model. The models with GP lower respiratory tract infections were better than the null model for every age band. Whilst in four out of seven of the age bands the smallest errors were for the GP upper respiratory tract infection models. Estimation of errors by season confirmed that 2017/18 had the least accurate forecasts for most models (Supplementary S1 Table).

The null models accurately predicted the long-term increasing trend in hospital admissions and day of the week and holiday effects. However, during seasons with higher than average syndromic activity the null model underestimated the peak hospital admissions. For example, towards the end of 2014 and 2017 syndromic models using GP consultations for upper respiratory tract infection provided higher forecasts than the underestimating null models (Fig 2).

## Discussion

### Main finding of this study

We found that seasonal peaks in syndromic respiratory indicators responded to changes in the timing of seasonal respiratory pathogen activity, whereas hospital emergency admissions consistently peaked on 29[th] or 30[th] December. Therefore, these findings suggest that syndromic data do not provide any additional early warning of the timing of the highest seasonal peak in hospital admissions each year. Furthermore, we found that predictable seasonal, secular and day of the week effects are evident in emergency hospital admissions. However, during atypical influenza seasons, such as 2014/15 and 2017/18, there were excess hospital admissions that were associated with increased activity seen in syndromic respiratory indicators. Therefore, syndromic data can improve real-time forecasts of the intensity of peaks in emergency hospital admissions and excess activity outside of the peak period associated with seasonal respiratory disease.

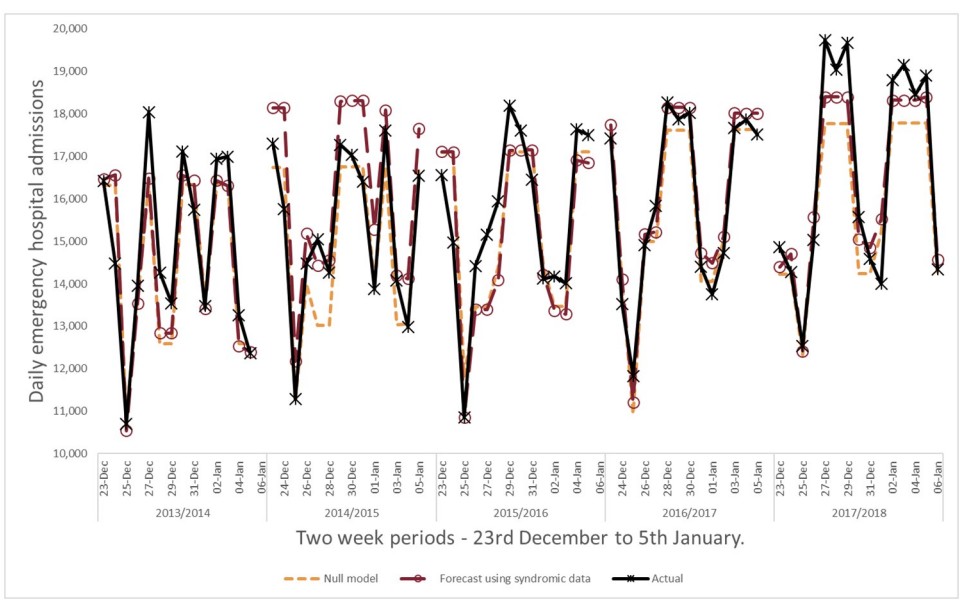

**Fig 2. Example forecast using GP consultations for upper respiratory tract infection compared to null forecast model with no syndromic data.**

## What is already known on this topic

In some countries, hospital admission data are available in real-time and are part of syndromic surveillance [18–20]. However, in countries like France and the United Kingdom (UK) full validated admission episodes including diagnostic data are not available until after discharge, and often not for several months post-discharge [21].

Many studies into the burden of seasonal respiratory disease on hospitals have focussed on admissions with a diagnosis of influenza. However, Fleming at al. studied all respiratory admissions because they found that only a small proportion of respiratory admissions received an influenza diagnosis, and that complications of influenza could result in later admissions for pneumonia etc. [22]. In our study, we aimed to show the utility of syndromic surveillance for assisting hospital planners manage total demand, and therefore we considered all emergency admissions, not just those with a respiratory diagnosis.

Previous studies into associations between syndromic systems and hospital admissions have used GP in-hours syndromic data. Specifically, associations have been found between GP ILI consultations and hospital admissions for influenza and pneumonia in the elderly [13, 22, 23]. In addition to ILI, GP consultations for acute bronchitis in the elderly have been found to be associated with hospital respiratory admissions, linked to RSV activity [1]. Studies exploring syndromic associations with admissions have produced estimates for the percentage of attributable variation in hospital admissions that vary from 2.7% [22] to 15% [13].

Comparisons between time series and forecasting studies naturally consider lags in terms of temporal association and whether trends in one series consistently anticipate trends in another. Chan et al. found that the greatest lag and therefore potential for early warning of changes in hospital admissions was from children aged 5–17 years with ILI [24], whilst, Schanzer et al. found that the relationship between ED ILI attendances and hospital admissions was not consistent enough for forecasting [25]. However, it is often more useful to planners to predict peaks when demand is at its greatest, rather than these temporal associations [26]. Therefore, our study focussed on the timing of peak demand for hospital admissions and the intensity

during these peaks. Future studies may also consider whether syndromic data can anticipate the start of surges in demand that lead to these peaks.

Previous studies in Canada and the UK have identified a consistent peak in hospital admissions between Christmas and New Year that leads to bed pressures and is independent of the timing of the peak in influenza incidence [1, 13, 23]. For example, Elliot et al. found that although the timing of peaks in GP ILI consultations varied widely by season, between 1990 and 2005 all but one of the 15 seasons had a peak in elderly respiratory hospital admissions occurring between weeks 52 and 02. Therefore, studies into associations between syndromic data and hospital admissions usually incorporate seasonality into their models [3], and seasonality is often found to have a stronger association than syndromic data [2, 13, 23, 26]. By contrast, Fuhrman et al. found that the variation in timing of peaks in hospital admissions in France was strongly associated with peak timing for GP ILI consultations [21]. However, this French study considered just hospital admissions for chronic obstructive pulmonary disease and not the totality of burden.

The timing of the peak in hospital admissions between Christmas and New Year complicates daily forecasting using syndromic data because public holidays affect both the availability of services and patient behaviours [27]. Elliot et al. imputed weekly rates during the holiday period from adjacent weeks [1]. By contrast, we smoothed the data to remove holiday effects, using methods for routine surveillance described elsewhere [16]. PHE uses a range of syndromic systems, including both GP in-hours and out-of-hours to provide surveillance coverage all year round, including holiday periods.

We found that hospital admission peak intensity was significantly associated with syndromic indicators, seasonality, and an increasing secular trend. Other studies also found a significant increasing secular trend in hospital admissions for elderly patients [1, 2, 13, 22]. Patients over 75 years have been found to account for most hospital influenza admissions and the greatest pressure on bed days [22].

## What this study adds

The findings from this study have supported earlier studies that showed an association between syndromic data and hospital admissions, and with the seasonality of hospital admissions being less sensitive to changing trends in seasonal respiratory disease than syndromic indicators. Furthermore, our results have confirmed that syndromic data can assist in predicting excess admissions associated with syndromic indicators for seasonal respiratory disease.

Many previous studies on the association between syndromic data and hospital admissions focus just on specific age groups or admissions with specific respiratory diagnoses. One of the strengths of our study is the consideration of all emergency admissions, which represent the excess demand that confront those managing health care pressures each winter. Furthermore, whilst previous studies have considered just one source of syndromic data, usually GP consultations, we have been able to use a range of syndromic data available from a working system, including telehealth calls, GP consultations and ED attendance data. Finally, unlike previous studies, we have studied daily not weekly data as any future forecasts developed from syndromic data will be most applicable if used in real-time.

## Limitations of this study

The data used in this study were not patient identifiable linked data and therefore a causal link between increases in respiratory illness, syndromic indicators and hospital admissions cannot be demonstrated. Also, we have not studied other potential causal factors for hospital

admissions, such as temperature or the timing of school holidays. Future work may be able to improve forecasts by including environmental data where it is available in real-time.

Our primary concern was to provide accurate forecasts; therefore, we have not presented an analysis of how the strength of associations between syndromic indicators and hospital admissions vary when a time lag is included. Future work could explore lags between time series to provide a deeper understanding of which syndromic indicators are most likely to provide early warning of hospital admissions.

For simplicity, we only assumed a linear secular trend in our regression models. All models underestimated the number of admissions in 2017/18 and it may be that there is a long-term increasing non-linear trend.

This study has only considered individual syndromic indicators monitored through real time systems. There may be further potential in exploring the combination of individual syndromic indicators, which may improve the accuracy of the forecast models.

Finally, whilst the timing of peaks in admissions are important, it would also be useful to give early warning of the surges in demand that lead to peaks. Future work could ascertain whether syndromic surveillance is able to predict any initial surges in hospital admissions.

## Implications for public health practice

We have shown that syndromic surveillance data can improve real-time forecasts of peak intensity for emergency hospital admissions. Furthermore, we have used syndromic data available daily as part of working surveillance systems. Therefore, our results demonstrate that real-time forecast models can be constructed which use syndromic data to improve accuracy during seasons when there are additional pressures due to above average seasonal respiratory illness.

## Supporting information

**S1 Table. Absolute mean forecast errors (daily admissions) in England by age band, stratified by season.**
(DOCX)

## Acknowledgments

We acknowledge support from NHS Digital and NHS 111; Royal College of Emergency Medicine emergency departments participating in the Emergency Department Syndromic Surveillance System (EDSSS); EMIS Health and L2S2 Ltd; out-of-hours providers submitting data to General Practitioners Out-of-Hours and Advanced; The Phoenix Partnership (TPP), and participating SystemOne practices and University of Oxford, ClinRisk, EMIS Health, and EMIS practices submitting data to the QSurveillance database.

## Author Contributions

**Conceptualization:** Andre Charlett, James Mapstone, Gillian E. Smith, Alex J. Elliot.

**Formal analysis:** Roger A. Morbey.

**Methodology:** Andre Charlett, James Mapstone, James Sedgwick, Gillian E. Smith, Alex J. Elliot.

**Supervision:** Andre Charlett, Alex J. Elliot.

**Validation:** James Mapstone, Richard Pebody, James Sedgwick, Gillian E. Smith.

**Writing – original draft:** Roger A. Morbey.

**Writing – review & editing:** Andre Charlett, Iain Lake, James Mapstone, Richard Pebody, James Sedgwick, Gillian E. Smith, Alex J. Elliot.

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
