## [Decision Letter · Decision Letter 0]

29 Oct 2019

PONE-D-19-24776

Can syndromic surveillance help forecast winter hospital bed pressures in England? - Using routine daily syndromic surveillance data to forecast the winter peak in demand for hospital beds.

PLOS ONE

Dear Dr Morbey,

Thank you for submitting your manuscript to PLOS ONE. After careful consideration, we feel that it has merit but does not fully meet PLOS ONE’s publication criteria as it currently stands. Therefore, we invite you to submit a revised version of the manuscript that addresses the points raised during the review process.

I would be happy if you can clarify on the points both reviewers have raised, with particular focus on the methodological section. Many thanks!

We would appreciate receiving your revised manuscript by Dec 13 2019 11:59PM. To enhance the reproducibility of your results, we recommend that if applicable you deposit your laboratory protocols in protocols.io, where a protocol can be assigned its own identifier (DOI) such that it can be cited independently in the future. For instructions see: http://journals.plos.org/plosone/s/submission-guidelines#loc-laboratory-protocols

We look forward to receiving your revised manuscript.

Kind regards,

Oliver Gruebner

Academic Editor

PLOS ONE

Journal Requirements:

2. Please note that PLOS ONE has specific guidelines on software sharing (http://journals.plos.org/plosone/s/materials-and-software-sharing#loc-sharing-software) for manuscripts whose main purpose is the description of a new software or software package. In this case, new software must conform to the Open Source Definition (https://opensource.org/docs/osd) and be deposited in an open software archive. Please see http://journals.plos.org/plosone/s/materials-and-software-sharing#loc-depositing-software for more information on depositing your software." 2) please ask the following, ping the Data team with follow-up: "Please amend your Data availability statement to outline how other researchers may access the data used in this study, for instance by providing a direct link/URL or contact details, including an email address or phone number, to the relevant authority where the data is kept. Please also ensure that the specific dataset is identified, or that the Methods section contains enough detail for another researcher to reproduce the dataset.

3. Please remove the 'Draft not for onward circulation' watermark from the background of the manuscript pages (page 1 onwards).

Reviewers' comments:

Reviewer's Responses to Questions

**Comments to the Author**

1. Is the manuscript technically sound, and do the data support the conclusions?

Reviewer #1: Partly

Reviewer #2: Yes

2. Has the statistical analysis been performed appropriately and rigorously? 

Reviewer #1: I Don't Know

Reviewer #2: Yes

3. Have the authors made all data underlying the findings in their manuscript fully available?

Reviewer #1: No

Reviewer #2: Yes

4. Is the manuscript presented in an intelligible fashion and written in standard English?

Reviewer #1: Yes

Reviewer #2: Yes

5. Review Comments to the Author

Reviewer #1: In this study, the authors have investigated whether the range of syndromic

surveillance data available in the England may help to predict the winter

demand for hospital beds. This bed demand is clearly influenced by the amount

of respiratory disease in the community, so these syndromic data may indeed be

particularly valuable for healthcare planning and this is potentially a very

valuable study. However, I'm afraid that the methods section doesn't provide

enough detail about the model design and evaluation for me to thoroughly

assess the results and the authors conclusions.

Major comments

1. It's unclear exactly what each model comprised. Regarding the peak timing

models, the authors state:

Models included a variable for linear trend, and Fourier variables to model

seasonal variation (15). In addition to variables for weekends and public

holidays, 25 December was also included as an additional binomial variable

to account for differing presenting behaviour at Christmas compared to

other public holidays (16). Each syndromic indicator was modelled

separately, stratified by age group.

If I have interpreted this correctly, for each of the syndromic indicators

there were 7 regressions models, one for each of the following age groups:

under 1 year, 1 to 4, 5 to 14, 15 to 44, 45 to 64, 65 to 74 and 75+ years.

Is this correct? If so, presumably these 7 models (for a single indicator)

did not necessarily share the same coefficients for the linear term, the

seasonal terms, and the weekend, public holiday, and Christmas Day binomial

variables? What were the coefficients for each model, and which seasonal

terms (sines and cosines) were included?

2. What criteria was used to fit each model? Am I correct in thinking that the

authors minimised the mean absolute error, as they used in a later section

to assess model forecasts?

3. Regarding the peak timing models, why wasn't a null model (i.e., without a

syndromic indicator) included for comparison? This would be very helpful

for interpreting the adjusted R-squared values listed in Table 2.

4. The same models were then used to "assess the utility of syndromic data for

forecasting" (and this time a null model was included). In this case, for

each year Y in turn, the models were fit to data from all years except Y.

From Figure 2 I gather that the daily bed demand on day D was predicted by

each model using the syndromic indicator data up to, and including, day D.

Is this correct? If so, these results should be called "nowcasts" instead

of "forecasts". If not, and the models were used to predict daily bed

demand for future days D+1 onward, Figure 2 should identify the forecasting

date(s) and include confidence intervals.

5. In evaluating the improvements in forecast performance, the authors report

the mean absolute error for each model and each age group in Table 3,

averaged over the five winters. Relative to the null model, some of the

syndromic models showed substantial improvements for children aged under 5

and adults aged 75+. But this was not evident for those aged 5-74. Did the

authors also look at, e.g., histograms of the absolute errors for each

model? It would be very interesting to know whether some of the models did

yield better predictions than the null model for those aged 5-74 for the

most part, but also yielded a handful of predictions with large errors.

6. It would also be very useful to see the mean absolute errors presented

separately for each year, since the study period (2013-2018) included two

"atypical influenza seasons" (2014/15 and 2017/18). This should hopefully

provide more detailed evidence to support the authors claim that "syndromic

data can improve real-time forecasts of the intensity of peaks in emergency

hospital admissions and excess activity outside of the peak period

associated with seasonal respiratory disease".

In a similar vein, how did the adjusted R-squared for each model (as listed

in Table) vary from one year to the next? This could be nicely presented in

a reasonably simple plot, and would complement the results in Table 2.

Minor comments

1. In the methods section, I find the sub-section titles "Peak timing" and

"Peak intensity" somewhat confusing, when they could seemingly be titled

"Model fitting" and "Model nowcasts/forecasts". This is especially true

since the peak demand for hospital beds was always the 29th or 30th of

December.

Reviewer #2: Thank you for an excellent manuscript. I have read it with great interest. The aims and methods of the study are clearly stated, results are presented in a clear way, figures illustrate the most important points, discussion offers relevant insights into the applicability of the results and future directions for research, and conclusion corresponds to the stated aims. I would only have very minor suggestions and questions for the authors to consider:

1. The conclusion, as formulated in the first paragraph of Discussion, outlines very well what is the biggest advantage of adding syndromic indicators (forecasting the intensity of peak activity in atypical seasons), and where it add less value (predicting timing of the highest seasonal peak). Perhaps the conclusion in the Abstract could be specified similarly.

2 The authors discuss why they used all emergency admissions rather than respiratory admissions as outcome (264-266). I wonder if the reasoning for this choice could be mentioned already in the Introduction. Could using all admissions contribute to the regular peak in 29-30 December? Potentially, respiratory-diseases related admissions' peak might be more variable.

3. Moving averages are described in Methods (line 110) and Figure 1 suggests that they are averaged over 7 days. If that is correct, this could be mentioned in the Methods.

4. The authors discuss briefly the lag of temporal association (from line 224). Have you also modelled the emergency admissions considering some time lag (the indicator value from x days ago)? As a reader, I would be interested to see whether you tested such models as well, and if not, why.

5. Lines 377 and 379 could be removed.

6. PLOS authors have the option to publish the peer review history of their article (what does this mean?). If published, this will include your full peer review and any attached files.

Reviewer #1: No

Reviewer #2: No

---

## [Author Response · Author response to Decision Letter 0]

6 Dec 2019

Response to reviewers

Thank you for the helpful comments from the editor and reviewers. We have addressed each point raised below, our responses shown in red.

Editor comments

Please remove the 'Draft not for onward circulation' watermark from the background of the manuscript pages (page 1 onwards).

We have removed the watermark from the manuscript.

Please amend either the title on the online submission form (via Edit Submission) or the title in the manuscript so that they are identical.

We have amended the title in the online submission form so that it is the same as in the manuscript.

Reviewers' comments:

Reviewer #1: 

In this study, the authors have investigated whether the range of syndromic surveillance data available in the England may help to predict the winter demand for hospital beds. This bed demand is clearly influenced by the amount of respiratory disease in the community, so these syndromic data may indeed be particularly valuable for healthcare planning and this is potentially a very valuable study. However, I'm afraid that the methods section doesn't provide enough detail about the model design and evaluation for me to thoroughly assess the results and the authors conclusions.

We thank the reviewer for their constructive comments throughout their comprehensive review of our paper. The major comments seem to be asking more for clarification for the reviewer rather than requesting amendments to the manuscript however where appropriate we have provided additional material to provide further clarification for the reader.

Major comments

1. It's unclear exactly what each model comprised. Regarding the peak timing models, the authors state:

Models included a variable for linear trend, and Fourier variables to model seasonal variation (15). In addition to variables for weekends and public holidays, 25 December was also included as an additional binomial variable to account for differing presenting behaviour at Christmas compared to

other public holidays (16). Each syndromic indicator was modelled separately, stratified by age group.

If I have interpreted this correctly, for each of the syndromic indicators there were 7 regressions models, one for each of the following age groups: under 1 year, 1 to 4, 5 to 14, 15 to 44, 45 to 64, 65 to 74 and 75+ years.

Is this correct? If so, presumably these 7 models (for a single indicator) did not necessarily share the same coefficients for the linear term, the seasonal terms, and the weekend, public holiday, and Christmas Day binomial variables? What were the coefficients for each model, and which seasonal

terms (sines and cosines) were included?

The reviewer is correct in their interpretation that there were separate regression models for each syndromic indicator, each of which with separate coefficients for the different terms, seasonal, weekend etc. We do not feel that it is possible to present separately the coefficients for each model because there are too many separate models to make this practicable. Furthermore, in addition to separate models for each indicator and age group we also stratified by four English regions and five seasons (we have not presented separate results by region because we felt this was an unnecessary complication that did not add to the paper). Whereas the stratification by season was necessary for the k-fold validation of forecasts. Consequently, to publish model coefficients would require an additional 1540 (=11*7*4*5) tables! 

Two seasonal terms were used in our models, the following code shows how they were created in Stata using the day of the year (doy) function: cos(2*_pi*doy(date)/365), sin(2*_pi*doy(date)/365) We have amended the text to clarify that a single pair of Fourier terms were used.

2. What criteria was used to fit each model? Am I correct in thinking that the authors minimised the mean absolute error, as they used in a later section to assess model forecasts?

Yes, this is correct in that we sought to minimise mean absolute error in our forecast model fitting.

3. Regarding the peak timing models, why wasn't a null model (i.e., without a syndromic indicator) included for comparison? This would be very helpful for interpreting the adjusted R-squared values listed in Table 2.

Thank you for this suggestion, we have added a row for a null model to table 2.

4. The same models were then used to "assess the utility of syndromic data for forecasting" (and this time a null model was included). In this case, for each year Y in turn, the models were fit to data from all years except Y. From Figure 2 I gather that the daily bed demand on day D was predicted by

each model using the syndromic indicator data up to, and including, day D. Is this correct? If so, these results should be called "nowcasts" instead of "forecasts". If not, and the models were used to predict daily bed demand for future days D+1 onward, Figure 2 should identify the forecasting

date(s) and include confidence intervals.

The reviewer is correct in saying daily bed demand on D was predicted using syndromic data available up to day D, however this does not include syndromic data for day D, only up to the day before. Our aim was to replicate the conditions in which forecasts could be applied within our surveillance service, i.e. a daily service where each day we are looking at the syndromic data collected during the previous day. We did not include confidence intervals for figure 2 because it is not in the format where the left most point represents the date of the forecast with a single fan-shaped confidence interval showing increasing uncertainty over time. Instead figure 2 shows a series of daily forecasts each predicting just one-day ahead.

5. In evaluating the improvements in forecast performance, the authors report the mean absolute error for each model and each age group in Table 3, averaged over the five winters. Relative to the null model, some of the syndromic models showed substantial improvements for children aged under 5 and adults aged 75+. But this was not evident for those aged 5-74. Did the authors also look at, e.g., histograms of the absolute errors for each model? It would be very interesting to know whether some of the models did yield better predictions than the null model for those aged 5-74 for the most part, but also yielded a handful of predictions with large errors.

We have added a line to the text to reflect that we did examine the histograms for the different indicators across the age bands and did not find any evidence of outliers having undue influence. Although, there was a wider variance and longer tails for the 5-64 years’ age bands this was also the case for the null model.

6. It would also be very useful to see the mean absolute errors presented separately for each year, since the study period (2013-2018) included two "atypical influenza seasons" (2014/15 and 2017/18). This should hopefully provide more detailed evidence to support the authors claim that "syndromic data can improve real-time forecasts of the intensity of peaks in emergency hospital admissions and excess activity outside of the peak period associated with seasonal respiratory disease".

We agree with the reviewer that showing mean absolute errors for each season could be useful so we’ve added this as a supplementary table, S1 and provided some additional text in the results. In most cases the absolute errors are highest for 2017/18 which agrees with our commentary that the models under-estimated this season.

In a similar vein, how did the adjusted R-squared for each model (as listed in Table) vary from one year to the next? This could be nicely presented in a reasonably simple plot, and would complement the results in Table 2. 

We decided not to include a table or plot of model fit by season because we felt this would require considerable extra explanation to ensure the results were not miss-interpreted. The reason these results could be miss-interpreted is due to the k-fold validation; e.g. the models we created for forecasting the 2017/2018 season included data from all 5 seasons except 2017/18, we didn’t create forecast models using just the data for 2017/2018. Therefore, if we were to present a version of table 2 stratified by season we’d have to explain that each model is based on the exclusion of one year instead of the inclusion of one year. Consequently, if season 2017/2018’s data was causing a poor fit this would result in all of the models having a poorer fit except for the one labelled “2017/2018 excluded.” 

Minor comments

1. In the methods section, I find the sub-section titles "Peak timing" and "Peak intensity" somewhat confusing, when they could seemingly be titled "Model fitting" and "Model nowcasts/forecasts". This is especially true since the peak demand for hospital beds was always the 29th or 30th of

December.

We agree with the reviewer that these sub-headings are unhelpful, particularly for the methods section, and we welcome the suggestion to change them to Model fitting and Model forecasts. For consistency, we have also changed the sub-heading in the results.

Reviewer #2: 

Thank you for an excellent manuscript. I have read it with great interest. The aims and methods of the study are clearly stated, results are presented in a clear way, figures illustrate the most important points, discussion offers relevant insights into the applicability of the results and future directions for research, and conclusion corresponds to the stated aims. I would only have very minor suggestions and questions for the authors to consider:

We thank the reviewer for their very positive and supportive review. 

1. The conclusion, as formulated in the first paragraph of Discussion, outlines very well what is the biggest advantage of adding syndromic indicators (forecasting the intensity of peak activity in atypical seasons), and where it add less value (predicting timing of the highest seasonal peak). Perhaps the conclusion in the Abstract could be specified similarly.

Thank you for this suggestion, we have changed the abstract conclusion accordingly.

2. The authors discuss why they used all emergency admissions rather than respiratory admissions as outcome (264-266). I wonder if the reasoning for this choice could be mentioned already in the Introduction. Could using all admissions contribute to the regular peak in 29-30 December? Potentially, respiratory-diseases related admissions' peak might be more variable.

Prior to our study, we did spend considerable time discussing with relevant experts the advantages and disadvantages of modelling ‘all emergency admissions’ vs ‘respiratory admissions’. Although, we believed that using just respiratory emergency admissions would give us stronger associations and better model fit, our stakeholders informed us that a forecast for total emergency admissions would be more valuable for healthcare service planners dealing with and managing pressures. Also, the greatest pressures felt by hospitals are when total emergency admissions peak, whatever the case mix. Therefore, we decided that the most useful research question was whether syndromic data was useful for forecasting all emergency admissions. We agree with the reviewer that this should be mentioned earlier in the paper and have added a sentence to the methods section under data collection.

3. Moving averages are described in Methods (line 110) and Figure 1 suggests that they are averaged over 7 days. If that is correct, this could be mentioned in the Methods.

We have added clarification to the methods that it is a 7-day moving average that is used.

4. The authors discuss briefly the lag of temporal association (from line 224). Have you also modelled the emergency admissions considering some time lag (the indicator value from x days ago)? As a reader, I would be interested to see whether you tested such models as well, and if not, why.

We did explore lags in association in the preparation work for this study. However, our primary concern was to create as accurate a forecast as possible rather than model the lagged associations between the time series and we found the most accurate forecasts included the most recent data available. We have added a comment to the limitations section to note the importance of considering time lags in any future studies.

5. Lines 377 and 379 could be removed.

 We have removed the un-necessary heading for Figure Legends.

[end]

---

## [Decision Letter · Decision Letter 1]

24 Jan 2020

Can syndromic surveillance help forecast winter hospital bed pressures in England?

PONE-D-19-24776R1

Dear Dr. Morbey,

We are pleased to inform you that your manuscript has been judged scientifically suitable for publication and will be formally accepted for publication once it complies with all outstanding technical requirements.

Please see the comments of reviewer #1 under point 6 below. Thank you!

With kind regards,

Oliver Gruebner

Academic Editor

PLOS ONE

Additional Editor Comments (optional):

Reviewers' comments:

Reviewer's Responses to Questions

**Comments to the Author**

1. If the authors have adequately addressed your comments raised in a previous round of review and you feel that this manuscript is now acceptable for publication, you may indicate that here to bypass the “Comments to the Author” section, enter your conflict of interest statement in the “Confidential to Editor” section, and submit your "Accept" recommendation.

Reviewer #1: (No Response)

Reviewer #2: All comments have been addressed

2. Is the manuscript technically sound, and do the data support the conclusions?

Reviewer #1: Yes

Reviewer #2: Yes

3. Has the statistical analysis been performed appropriately and rigorously? 

Reviewer #1: Yes

Reviewer #2: Yes

4. Have the authors made all data underlying the findings in their manuscript fully available?

Reviewer #1: No

Reviewer #2: Yes

5. Is the manuscript presented in an intelligible fashion and written in standard English?

Reviewer #1: Yes

Reviewer #2: Yes

6. Review Comments to the Author

Reviewer #1: In this study, the authors have investigated whether the range of syndromic

surveillance data available in the England may help to predict the winter

demand for hospital beds. This bed demand is clearly influenced by the amount

of respiratory disease in the community, so these syndromic data may indeed be

particularly valuable for healthcare planning and this is potentially a very

valuable study.

I thank the authors for their responses to my original comments, which have

thoroughly addressed my concerns about the level of detail in the methods

section. I have only one minor suggestion (see note 3, below) about this

revised version.

1. Given the multiple levels of model stratification, I agree with the authors

that the sheer number of model parameters is simply too large to include as

additional tables. Thank you for clarifying that each model included two

seasonal terms, rather than an arbitrary number of seasonal terms.

2. Thank you for including null model results in Table 2, which I find helpful

for putting the results obtained from the other models into context.

3. My confusion about forecasting versus nowcasting stems from the use of the

phrase "[using data] up to day D", because I find this wording ambiguous

about whether day D itself is included. I think it would be helpful to

include a remark to this effect in the "Model forecasts" part of the

Methods section.

4. I thank the authors for noting in the text that there was no evidence of

outliers affecting the mean absolute errors.

Reviewer #2: (No Response)

7. PLOS authors have the option to publish the peer review history of their article (what does this mean?). If published, this will include your full peer review and any attached files.

Reviewer #1: No

Reviewer #2: No

---

## [Editor Report · Acceptance letter]

28 Jan 2020

PONE-D-19-24776R1 

Can syndromic surveillance help forecast winter hospital bed pressures in England? 

Dear Dr. Morbey:

I am pleased to inform you that your manuscript has been deemed suitable for publication in PLOS ONE. Congratulations! Your manuscript is now with our production department. 

With kind regards,

on behalf of

Dr. Oliver Gruebner 

Academic Editor

PLOS ONE